# Semantic Variant Primary Progressive Aphasia: Practical Recommendations for Treatment from 20 Years of Behavioural Research

**DOI:** 10.3390/brainsci11121552

**Published:** 2021-11-23

**Authors:** Aida Suárez-González, Sharon A. Savage, Nathalie Bier, Maya L. Henry, Regina Jokel, Lyndsey Nickels, Cathleen Taylor-Rubin

**Affiliations:** 1Dementia Research Centre, UCL Queen Square Institute of Neurology, University College London, London WC1N 3BG, UK; 2School of Psychological Sciences, College of Engineering, Science and Environment, The University of Newcastle, Callaghan, NSW 2308, Australia; sharon.savage@newcastle.edu.au; 3Centre de Recherche de l’Institut Universitaire de Gériatrie de Montréal, Montreal, QC H3W 1W4, Canada; nathalie.bier@umontreal.ca; 4School of Rehabilitation, Faculty of Medicine, Université de Montréal, Montreal, QC HEC 3J7, Canada; 5Department of Speech, Language, and Hearing Sciences, The University of Texas at Austin, Austin, TX 2504A, USA; maya.henry@austin.utexas.edu; 6Dell Medical School, The University of Texas at Austin, Austin, TX 78712, USA; 7Rotman Research Institute, Baycrest Health Sciences, Toronto, ON M6A 2E1, Canada; rjokel@research.baycrest.org; 8Department of Speech-Language Pathology, University of Toronto, Toronto, ON M5G 1V7, Canada; 9School of Psychological Sciences, Macquarie University, Sydney, NSW 2109, Australia; lyndsey.nickels@mq.edu.au (L.N.); Cathleen.Taylor@health.nsw.gov.au (C.T.-R.); 10Department of Speech Pathology, Uniting War Memorial Hospital, South Eastern Sydney Local Health District War Memorial Hospital, Waverley, NSW 2024, Australia

**Keywords:** semantic dementia, semantic variant primary progressive aphasia, word finding, frontotemporal dementia, language therapy, behavioural therapy

## Abstract

People with semantic variant primary progressive aphasia (svPPA) present with a characteristic progressive breakdown of semantic knowledge. There are currently no pharmacological interventions to cure or slow svPPA, but promising behavioural approaches are increasingly reported. This article offers an overview of the last two decades of research into interventions to support language in people with svPPA including recommendations for clinical practice and future research based on the best available evidence. We offer a lay summary in English, Spanish and French for education and dissemination purposes. This paper discusses the implications of right- versus left-predominant atrophy in svPPA, which naming therapies offer the best outcomes and how to capitalise on preserved long-term memory systems. Current knowledge regarding the maintenance and generalisation of language therapy gains is described in detail along with the development of compensatory approaches and educational and support group programmes. It is concluded that there is evidence to support an integrative framework of treatment and care as best practice for svPPA. Such an approach should combine rehabilitation interventions addressing the language impairment, compensatory approaches to support activities of daily living and provision of education and support within the context of dementia.

## 1. Introduction

In the 1970s, Warrington’s description of three individuals with a selective and profound inability to name and recognise objects [1] laid the foundation for what years later, in 1989, would be coined “semantic dementia” [2]. Semantic dementia, now widely referred to as semantic variant primary progressive aphasia (svPPA), is a neurodegenerative syndrome characterised by progressive loss of semantic knowledge in the context of otherwise well-preserved language and cognitive abilities [3,4]. Current consensus criteria require language impairment to be the most salient clinical symptom and the main cause of impairment in daily living activities [3,5]. Clinically, individuals with svPPA present with fluent speech (preserved repetition and speech production) and loss of semantic knowledge across all modalities of testing (e.g., picture naming, single-word comprehension and visual association tasks). As the disease progresses, behavioural features emerge, and speech becomes increasingly empty, culminating with mutism in the final stages [6]. An illustrative example is provided by the response of one woman with svPPA who, when asked about her symptoms, pointed to the trees in the hospital’s courtyard and said, “I don’t know what those green things are anymore”.

SvPPA is estimated to account for one-third of all cases of frontotemporal dementia [7] with an average age at symptom onset of 60 years (64 years for diagnosis to be established). The prognosis for length of survival following diagnosis is highly variable, with a median of 12 years [8]. MRI brain scans typically reveal bilateral and asymmetric temporal pole atrophy (greater on the left) and asymmetric anterior hippocampal atrophy [9]. Furthermore, the anterior portion of the fusiform gyrus and adjacent regions are also critical areas systematically affected in svPPA and appear to play a pivotal role in semantic degradation [10,11,12,13]. Between 75% and 100% of all svPPA cases are associated with underlying TDP-43-C pathology, with the remainder mostly involving FTD tau [8,14,15,16] and a small proportion of cases showing concomitant Alzheimer’s disease pathology [8,17].

There is no curative or disease-modifying treatment for svPPA. However, a growing body of research on non-pharmacological interventions has shown that people with svPPA may relearn lost vocabulary and benefit from other behavioural therapies. The first rehabilitation reports emerged in the literature in the late 1990s, inspired by patients who spontaneously engaged in self-practice as an attempted remedy for their anomia [18,19]. The proliferation of single case studies and small group studies over the next decades have demonstrated that people with svPPA who receive naming therapy can improve their recall of object labels in the short term, that the gains might be retained over time and that at least partial restoration of semantic knowledge may be possible (see reviews by Carthery-Gouland et al. [20], Jokel et al., [21], Cotelli et al., [22] and Pagnoni et al. [23] for an overview). Furthermore, the breadth of research into non-pharmacological interventions has by no means remained restricted to word retrieval. Therapeutic approaches targeting conversation [24], tasks and activities of daily living [25,26,27], psychoeducation programmes [28,29] and peer support groups [30] have made headway and are on the increase. Altogether they have set the stage for an integrative framework of clinical treatment and care in svPPA that combines rehabilitation interventions, compensatory approaches and provision of education and support, addressing the language impairment in svPPA within the context of dementia [31]. This article aimed to synthesise the learnings from 20 years of research in the non-pharmacological treatment and management of svPPA and lay out evidence-based recommendations for clinical practice and future research. For the purposes of education and dissemination beyond an academic audience, this article includes a lay summary available in English, Spanish and French (Appendix A).

## 2. Anomia in svPPA as a Sign of Semantic Breakdown

There is evidence that the anomia seen in svPPA stems from impairment in semantic knowledge [32]. This is different from the word retrieval impairments shown in the other PPA variants that arise at the lexical/phonological (logopenic variant PPA) or post-lexical (non-fluent/agrammatic PPA) [33] stages. A basic understanding of how semantic memory architecture works is therefore required to develop effective treatments. A common theory is that semantic knowledge is organised in a hierarchy of specificity [1,34], ranging from very specific attributes at the bottom (e.g., the hummingbird is a small bird that can hover) to very general knowledge at the top (e.g., a hummingbird is an animal) (see Figure 1).

Specific attributes are hypothesised to degrade first in a continuum of progressive degeneration that continues with the loss of general attributes and culminates in the disappearance of the concept. For instance, a person may identify a hummingbird as a living thing without being able to identify its specific properties (e.g., that it can fly and feeds on flower’s nectar). This means that, during cognitive and language assessment, partial provision of information should not be interpreted as unequivocal proof of complete semantic preservation. Further investigation of semantic integrity should always be pursued in people with svPPA in preparation for therapy.

### Leveraging Episodic Memory

Episodic memory (e.g., the ability to remember where you parked your car, what you did yesterday evening, or the plumage of a bird that is new to you) is a main entry point of semantic information into the memory system. This new information is integrated into existing bodies of knowledge by a dual system supported by the hippocampus (allowing quick capture of episodes) and neocortical structures (allowing a slower but effective integration into a long-term database) [35]. More specifically, in this second neocortical stage, information is consolidated in integrated, generalisable representations across a network distributed along the neocortex, tapping into the sensory, motor and linguistic systems [36]. Cross-modal interaction of these areas has been hypothesised to be anatomically supported by the anterior temporal lobes (ATLs) that operate as a hub where different forms of semantic information converge and connect [37,38]. This ATL region is affected at an early stage by the bilateral pathological aggregation of proteins associated with svPPA. However, the brain structures supporting episodic memory, such as the posterior area of the hippocampus and posterior cingulate cortex [39], are usually reasonably preserved. This suggests that, in principle, the episodic memory gateway to inputs that will eventually transform into re-learned concepts may remain functional. Consequently, this mechanism may be used, in conjunction with partially degraded neocortical structures, to the advantage of rehabilitation goals [40].

## 3. Differences between Left and Right Variants: Implications for Practice

The usual pattern of brain atrophy in svPPA (left greater than right) is reversed in approximately 30% of cases (i.e., right greater than left), giving rise to left and right-sided variants (left-svPPA and right-svPPA respectively) [41,42,43] (see Figure 2). Left-svPPA is characterised by poorer performance on verbal tasks compared to right-svPPA [13,43,44]. At the time of presentation, the prevalence of word-finding difficulties in left-svPPA is reported to be 94%, compared to 36% in right-svPPA, while impairments in single-word comprehension are reported in 67% of left-svPPA and 18% of right [43]. In contrast, individuals with right-svPPA show greater impairment of non-verbal semantics [38,42,43]. In up to 91% of cases with right-svPPA, the clinical picture is characterised by prosopagnosia (a difficulty in recognising faces) that for these individuals is associated with person-specific semantic knowledge breakdown [42,45,46,47,48,49]. Behavioural changes, although reported in both variants, seem to be more pronounced and appear earlier in right-svPPA, with social awkwardness and loss of insight are commonly reported (present in 64% and 55% of individuals respectively) [43] along with loss of empathy, disinhibition, apathy and compulsiveness [42,45,48].

Analysis of the types of naming errors produced by each group suggests that individuals with right-svPPA might have more difficulty accessing semantic knowledge through visual than verbal modalities (e.g., more difficulty recognising a famous face by looking at a photograph than by listening to a description of the person) (see Table 1). Individuals with left-svPPA show a larger proportion of circumlocutions in response to naming difficulties (e.g., “when it rains” for umbrella) and omissions [44,50] compared to right-svPPA, while those with right-svPPA make more coordinate and superordinate semantic errors (e.g., coordinate: “cat” for “dog” and superordinate: “animal” for “dog”) [44,50]. The reduced ability of these individuals to access knowledge through visual features has been proposed as a possible mechanism that contributes to their greater difficulty in producing semantic associations, predisposing them to production of more taxonomic (coordinated and superordinate) semantic errors [44].

In light of this evidence, clinicians should pay particular attention to a few factors. First, whether verbal material (e.g., audio recordings, verbal descriptions and sounds) may be preferable to visual (e.g., photographs and real objects) should be considered when treating individuals with right-svPPA. Second, individuals with left-svPPA seem better able to access residual associated semantic knowledge and use this to describe the target when attempting to name. This can be used as a therapeutic opportunity, for instance, by encouraging the individual to retrieve this residual knowledge and relink it with the label.

## 4. The Current Evidence Informing Treatment and Management of Anomia and Word Comprehension Deficits

### 4.1. How Should Therapies for Anomia Be Designed and Administered?

Typically, lexical training therapies have consisted of a set of items given to individuals to practice. Therapy in svPPA should focus on maintaining or improving access to both names and semantic representations. Below, we present a summary of how these therapies should be planned and administered in svPPA based on a synthesis of current evidence.

### 4.2. Who Benefits from Anomia Therapy?

Benefits of therapy have been shown across a range of severities of anomia, provided some level of spoken language is preserved (i.e., there are no studies of individuals who are mute). This suggests that, in principle, the level of severity should not prevent any individual with svPPA from being considered for treatment, although the nature of the intervention would differ based on the level of severity. People in the early stages may have the advantage of retaining more semantic knowledge on which to build the therapy. They are also more likely to be free of other cognitive or behavioural symptoms that may impact successful engagement with therapy and, in fact, circumscribed semantic impairments longer than 6 years post-onset have been reported in some individuals [51,52,53,54].

### 4.3. How Many Sessions, of What Length and How Many Items per Session?

Current evidence suggests that 20–60 min of daily (or almost daily) practice is effective to produce short-term benefits [51,52,55,56,57], although some individuals have also shown benefits from less. Significant improvements should be expected within the first month of consistent practice [40,52,53,58,59,60,61,62,63,64] but may be evident sooner. Most studies to date have combined face-to-face sessions with the therapist with self-administered home programmes. Usual set size is between 15 and 30 items per session [51,52,53,54,58,60,63,65].

### 4.4. What Kind of Items and Naming Therapy?

Two kinds of words have been targeted in therapy: those that still are associated with some residual semantic knowledge and those that are not. A word is considered to have residual knowledge when the person can produce or comprehend at least partial information about it (e.g., “it’s food” for an egg, without being able to connect the association between an egg and a hen). These words are by far the most investigated in the lexical retrieval literature. Words where meaning is completely lost have, on the contrary, been less investigated and the few studies looking at the use of conceptual enrichment therapies to treat words destitute of semantic knowledge have produced mixed results [66,67]. A list of the techniques used in the svPPA rehabilitation literature is shown in Table 2.

One of the most common approaches to improving naming is the “Look, listen and repeat” (LLR) or “Repetition (and reading) in the presence of the picture (RRIPP)”. A picture of the target concept is presented, along with the name as a spoken and/or written word for the individual to repeat/read aloud, sometimes preceded by an attempt at naming, with or without (semantic or phonological) cues. Multiple variations of this approach have proven effective for improving production of vocabulary that the person with svPPA can still comprehend (see Table 2). However, this technique can lead to rote-learning (rigid and context-specific) and poor generalisation when semantic knowledge of the trained item is very impaired (e.g., the person can no longer comprehend either the lexical label or a picture of the object). Restitutive training of words/concepts that the person can no longer comprehend has been less explored in the literature. The suitability of a semantic approach to treating these items (e.g., working on characteristics of an object’s usage and location and linking it with other related memories) is supported by two types of studies. The first consists of studies looking at the direct restoration of semantic knowledge [66,67,70] and the second capitalising on residual semantic information to boost word retrieval [18,40,52,53,55,58,60,62,63,66,68,69,70,71]. Both contribute to understanding the importance of the semantic system in the rehabilitation of svPPA. For instance, the naming of items with residual semantic knowledge appears to be easier to rehabilitate than that of items completely devoid of meaning [53,58]. Likewise, greater success is achieved with familiar items—familiar concepts degrade slower due to the larger and stronger network of semantic connections that are regularly reinforced with use (e.g., the concept of a toaster, used daily for breakfast, will be retained for longer than a hammer that is borrowed from a neighbour and used occasionally) [53,58,60,84]. In this same vein, some authors have introduced photos of individuals’ own items within their therapy material (rather than generic exemplars), to harness familiarity and personal significance [51,54,65,82]. Others have identified semantic attributes of exemplars [76] or sorted items within semantic categories [18,51,54,65] to further reinforce the semantic concept (but randomise the order of items with each presentation to avoid rote learning).

### 4.5. Are These Therapies Well Accepted by People with svPPA?

Most studies of word retrieval therapy in svPPA have shown good adherence of people to practice. In many cases, participants completed home programmes consistently for many months. The first lexical retrieval therapy studies were prompted by individuals who started self-practice on their own initiative, evidencing their keenness to play an active role in their treatment [18]. Inevitably, individuals reported in the literature are those who volunteered for research and are probably particularly motivated to pursue therapy, which may not be the case when extrapolating to the broader clinical population. It has been reported that, in clinical settings, individuals with PPA who receive lexical retrieval therapy show a rate of adherence of 60% [85]. The authors of that study found that adherence was more likely when the treatment commenced in the year after diagnosis and when the patient was motivated, and mood was stable. Clearly, there will be people with svPPA who may prefer not to engage in lexical therapy for various reasons. In these cases, there is still a wide range of therapeutic options that can be offered (e.g., use of compensatory techniques, environmental adaptations, partner training and psychological support).

### 4.6. Are People with svPPA Aware of Their Deficits?

People with svPPA typically recognise that their language performance has weakened. However, some individuals appear to have difficulties evaluating their past knowledge of words (even in realising that certain words ever existed) and the extent of the impoverishment of their language content. For instance, Savage et al. [86] reported that people with svPPA who have mild to moderate semantic impairments showed no awareness of obvious mislabelling errors when naming components of objects. The authors of the study warn about the implications that this may have regarding patients’ role and input into rehabilitation planning and recommend that rehabilitation programmes should not be based on patients’ judgment alone and instead also involve family members and friends.

### 4.7. How Long Does the Effect of Therapy Last?

Many studies have demonstrated that the significant improvements in naming are often very well maintained over the first month after ceasing practice [40,52,54,56,59,63,81]. Outcomes beyond this, however, are variable. For some people with svPPA, a high proportion (73–82%) of the words named at the end of treatment can still be successfully named 3 to 6 months later [54,60,63,82]. In others, levels of retention in that time window are modest (e.g., around 65% of trained words) [53,62,65] or low (e.g., only 10–40% of words are maintained) [58,68]. Encouragingly, the majority of studies report performance that continues to be above baseline levels for up to 6 months after completing treatment [87]. These benefits have also been observed 12 months post-treatment in a small number of studies [82].

The extent of retention may be influenced by the degree of semantic knowledge still retained for an item (i.e., meaningful items persist longer [58]) and the opportunity to continue rehearsing items in everyday life [54,60,68]. This is consistent with observations that autobiographical experience and subsequent conversations regarding such experiences, may enhance semantic knowledge and preserve these words over time [81,82,84]. While this integration of the use of words in everyday life plays an important role in retaining vocabulary, many words (e.g., stove, plate) may not be used often enough in everyday conversation to allow regular practice, requiring alternative strategies for ongoing reinforcement. One feasible alternative is maintaining regular revision of the re-learned words. While daily practice may be needed in the early phases of an intervention, successful maintenance revisions (to maintain at least 80% of therapy items) require less practice [54]. For instance, when monitored over a 6 month period, people with svPPA with a moderate level of impairment needed less than 10 revision sessions over 6 months to maintain their naming. For those with more severe semantic impairment, Savage et al. [51,54] found that regular, weekly practice was needed to restore the benefits of the initial intense training. In particular, performance at around 2 months post-intervention appears to be a useful indicator of the frequency of revision that could be required for sustained maintenance—implying that this is a useful time point for clinicians to monitor and then formulate the revision programme for those people with mild to moderate svPPA.

A practical consideration for people with svPPA and their families then becomes how long to continue with interventions. In some cases [54,68], the practice simply becomes part of the usual routine or there may be enjoyment gained from it. Consistent with this, some studies have reported ongoing practice persisting for 1–2 years [55,88]. For some individuals, however, where declines in performance may become upsetting or practice becomes stressful, it may not be desirable to continue. In these circumstances, individuals with PPA and their families should be prepared for declines to emerge over the months that follow.

### 4.8. Does This Learning Generalise?

An important aspect of any rehabilitation programme is the degree to which improvements extend from the intervention to assist the person in their daily living. The generalisation of benefits in svPPA has been usually evaluated in two ways: (1) whether naming improvements extend from trained to untrained words and (2) whether words can be used by the person with svPPA in contexts that differ from the training format. Generalisation of naming improvements, extending from trained to untrained words, have been observed in some individuals with non-progressive aphasia, but usually only when the impairment is one of phonological encoding, in the absence of significant semantic or lexical deficits [89]. A consistent finding across most svPPA treatment studies is that untrained words do not improve [21,25] with very few exceptions showing the opposite result [81,82].

An alternative way of considering the generalisation of naming therapies is to evaluate the extent to which trained words can be used by the person with svPPA in contexts that differ from the training format. Broadly, this may be divided into “near transfer”—wherein the demonstration of knowledge is highly similar to the original training context (e.g., asking the person to produce the word in response to a different exemplar of the stimulus—see Figure 2 in Heredia et al. [60]) or “far transfer”—where knowledge must be applied more flexibly (e.g., by completing a different kind of language task such as verbal comprehension) [90]. Successful naming has been observed when people with svPPA are tested on alternative versions of trained items [52,63] or photographs of target items taken from different views [60,66] but much less when they are required to name visually dissimilar versions of the trained item [40,60,72,91]. Encouragingly, evidence of producing trained words in other contexts after word training, such as fluency tasks (in one individual [63], naming to description [66], describing short videos of everyday scenes [51] or in production of a simple sentence construction, have also been observed [91]. 

To increase the chances of people with svPPA being able to correctly use the trained words in their everyday lives, it is helpful to tailor training stimuli to visually match the objects found within a person’s home (likewise in actual object use, which was found to depend on personal familiarity with object exemplars [92]). 

### 4.9. What Evidence Do We Have about Prophylactic Treatment in svPPA?

Prophylactic/preventative treatment aims to help retain current abilities by practising intact skills or items. There is some evidence suggesting that such preventative interventions may hold value in svPPA [73,78]. Several studies have found that treatment of items that could already be successfully named may slow the progression of semantic loss and anomia for those items [52,53,54,56].

### 4.10. Can We Deliver These Therapies Remotely? What Evidence Do We Have?

Digital technologies in treatment programmes provide opportunities to increase access for those with svPPA and their families who struggle to access expert care because of geographic location. Delivery of treatment via telehealth is highly relevant given the limited access to services for many individuals with PPA [78,83,93,94]. Significant improvements in word retrieval have been achieved after completing home-based programmes using either hardcopy or computer-mediated materials [18,40,53,54,58,60,65,66,68,95]. Rogalski and colleagues examined the feasibility of teletherapy for 28 individuals with PPA [96] showing that treatment delivered via video conferencing has the potential to improve access to care for people with PPA. Two studies conducted on people with svPPA show that lexical retrieval therapy can be delivered in-person or by teletherapy with similar results [78,83].

### 4.11. What Are the Barriers and Facilitators of Online Therapy?

Recognition of the barriers to, and facilitators of, successful implementation of remote digital therapy, however, is extremely important in both the research and clinical setting. Disease severity has been noted by several studies to be a contraindication for remote therapy and there is a recommendation that individuals participating in remote therapy should preferably be in the early to mid-stages of disease progression [83,91,96]. The inherent requirements of a technology can also be a barrier with the quality of audio and the stability of the internet connection being a prerequisite to successful participation online. In addition, the individual must possess adequate computer skills or a suitable support person to facilitate participation, particularly when carrying out intervention independently at home rather than supervised over the internet.

An example of these barriers, acting in concert, is provided by Taylor-Rubin et al. [91], reporting a series of single-case design treatment studies where lexical retrieval treatment was delivered via a computer-mediated home programme. Two of three svPPA participants had significant improvement in verb and noun production, following lexical retrieval treatment. However, a third participant, Nsv, showed only marginal gains over two blocks of lexical retrieval treatment. The authors hypothesised that as Nsv was five years post-onset, the severity of impairment may have contributed to less positive treatment results. Practice logs indicated poor adherence with computer operating difficulties preventing completion of all treatment schedule sessions in the second block of treatment [91].

A further barrier can be the lack of contact with the therapist. Caregivers in Rogalski’s study reported that less than optimum opportunities for face-to-face support for the person with PPA, in times of distress, was a limitation of participation in the web-based treatment programme [96]. Similarly, caregivers of people with PPA, including svPPA, reported, in a study of treatment adherence, that home treatment programmes can be lonely and socially isolating and this would be anticipated to reduce adherence, “It is easier to fall off the wagon with a programme at home” [85]. Finally, the barrier of social isolation could be minimised by innovative networking; pairing peers with svPPA in small online groups, thus incorporating support, increased social participation and positive experiences [30].

## 5. Compensatory Approaches to Support Communication in svPPA

Aside from the direct treatment of language, a number of single-case studies have explored the benefit of using a compensatory approach to support language difficulties, particularly naming, in svPPA [26,97,98,99]. Compensatory approaches include the use of external devices to support communication, such as compensatory augmentative and alternative communication (AAC) systems [100]. These can be based on low (e.g., paper communication board or notebooks) or high technology (e.g., smartphones or tablets and computers); people with svPPA may use them in conjunction with verbal communication in a multi-modal way, multiplying the communication options available to them [100].

In two case studies, Bier and collaborators [26,97] explored the potential of using smartphone applications to help two people with svPPA learn how to search for information related to lost concepts through Internet search engines or a visual dictionary application named ARCUS^©^. This application aimed to support the retrieval of people’s names from a virtual name directory using clues or information chosen by the person with svPPA. ARCUS^©^ was successfully used by ND, a recreation therapist in a senior living facility with early svPPA [97]. In his work, ND had to identify a large number of people by name each day. At the start of the study, he used a paper notebook to do this, organised into several columns, each linked to a different piece of information (e.g., resident’s room number or employee’s job type). The authors converted ND’s notebook into a smartphone application to ease its use and reduce the stigma associated with it. ND phased out his paper-based compensatory system in favour of this new, more flexible name retrieval system. Four years later, ND had extended the use of ARCUS^©^ by adapting it to record information about grocery stores and food items to buy before he went shopping.

Another recent study has combined the classical use of mobile technology to develop CoChat, an app constructed on natural language processing (NLP) features, social media use, and just-in-time principles that was tested in two people with svPPA [98]. In this app the user takes a photograph with the tablet’s built-in camera, shares the pictures with the person’s simulated social network (e.g., family and friends) and sees comments to the images in real time. Results suggest that CoChat may improve word retrieval in a natural conversational context making conversations easier when using the app. As AAC devices and systems are becoming common practice in aphasia, further studies will have to deepen our understanding of how these types of tools can be optimised in svPPA.

Semantic deficits may sometimes prevent people with svPPA from understanding task requirements and limit their ability to learn certain functions of assistive technological devices [26,97] (e.g., being able to remember the series of actions required to obtain an Internet connection, but not understanding why). Nevertheless, taken together, these case studies suggest that it is possible to teach the use of practical, portable solutions to compensate for semantic memory deficits. Considering the degenerative nature of svPPA, it is important to integrate AAC with other treatment approaches as early as possible in the disease process so that they are well practised before the skills to acquire their use are lost [98,100]. Finally, although strategies of functional communication have been explored in individuals with PPA in general, there is a lack of studies examining non-AAC compensatory strategies targeting svPPA in particular.

## 6. Interventions to Support Activities of Daily Living

Complementary approaches that support engagement or re-engagement in meaningful activities of daily living are also important. Participation in meaningful activity is the primary focus of these kinds of interventions, without special consideration for language skills—although these may also benefit. They are oriented toward two objectives: (1) capitalising upon preserved episodic (e.g., what you had for lunch) and procedural memory functions (e.g., how to perform different skills, such as tying your shoes); (2) focusing on significant and meaningful everyday activities that will have immediate results and a potential impact on well-being.

To our knowledge, only two studies have explored the engagement, or re-engagement, in meaningful daily living activities in svPPA [27,101]. In the first study, Bier et al. [27] did so by combining the repeated practice of an activity that the person had stopped doing (e.g., meal preparation) with a step-by-step cognitive assistive technology (SemAssist^©^). The objective was to support EC, a woman with left svPPA, to relearn how to prepare a specific recipe of her choice. This study showed that EC mainly used SemAssist^©^ to follow the current steps during the activity. While she made many mistakes before the therapy sessions began, she was able to complete the recipe without error by the end of the process. Interestingly, EC also resumed spontaneous preparation of other recipes, showing that she had acquired new “knowledge” about the ingredients from the recipes she practised (e.g., “goes in the shrimp recipe”) and did not overgeneralise. In the second study, O’Connor et al. [101] applied the Tailored Activity Program (TAP) with a person with svPPA who had highly repetitive routine behaviours. The TAP intervention resulted in this person engaging well in prescribed activities, with scores reflecting reduced carer distress regarding challenging behaviours and improved caregiver vigilance.

It therefore seems appropriate and promising to combine traditional language-based approaches with an interdisciplinary intervention that also incorporates a participatory approach such as occupational therapy or other meaningful activity interventions in svPPA.

## 7. Support Groups and Educational Programmes

One of the most recent developments in therapy for primary progressive aphasia is in group-based programmes offering education and support. While none of the published reports are specific to a particular PPA variant, they do include individuals with svPPA. In 2017, Jokel and colleagues published the first report of a group intervention programme that included both individuals with PPA and their caregivers [102]. The group members not only shared the intervention focus but, importantly, actively participated in defining it. Half of each session was spent on education, counselling and/or training communication strategies in dyads. The other half was separated into language activities for people with PPA and networking activities for caregivers. All participants reported valuing learning about the nature, progression and types of PPA, becoming familiar with current research in PPA, and several other aspects of the intervention. Components that were reported to be beneficial included receiving information on nutrition and lifestyle to support brain health, learning strategies for managing stress and depression, feeling understood by others in the group when experiencing difficulties during verbal communication, and getting support from multiple disciplines.

Although not specific to svPPA, to date, three more group interventions for PPA have also been reported [29,30,98]. Mooney and colleagues [98] developed a PPA group treatment model that incorporated elements of three methodologies used in language rehabilitation: communication strategies from augmentative and alternative communication, communication partner training from aphasia rehabilitation, and systematic instruction from dementia management. Morhardt et al. [29] describe the development of a programme that offered education, communication strategies, strategies to “live well” with PPA and non-language-based activities (e.g., watercolour painting and horticultural therapy). Finally, Taylor-Rubin et al. [30] delivered PPA education and support for a group of people with PPA and their caregivers in the early post-diagnostic period. In the post-intervention interview, participants highlighted the reduced feelings of isolation, increased feelings of support, increased knowledge of coping strategies and improved understanding of PPA as a result of participating in the programme soon after the receipt of the PPA diagnosis.

Based on the outcomes of these group interventions in PPA, several factors have emerged that may be critical to PPA care. First and foremost, the needs of both patients and caregivers should be addressed, preferably simultaneously [28,30]. A successful intervention programme for PPA should provide not just language activities and education but also a safe forum for discussing important and difficult issues, for sharing successes and failures, and for peer education. Such a programme is likely to ultimately result in improvements in confidence and well-being for both individuals with PPA and their caregivers. Published studies underline that PPA-specific education and ecologically valid context (i.e., group format) are positive elements highlighted by all participants. In addition, having consistent peer support helps to “normalise” daily challenges. It has been suggested that self-help groups may be beneficial in maintaining the group intervention benefits and they are recommended even in the absence of professional input [103].

As more and more services are being offered online, the support for individuals dealing with svPPA may also migrate to virtual space. A review of virtual support groups for dementia caregivers [104] suggests that weekly or monthly sessions can provide participants with knowledge about dementia, caregiving skills, coping strategies and access to resources. While occasional barriers, such as technology and access, were identified, there are also numerous economic and geographical advantages to online group sessions. Extrapolating from the broad dementia field, we may predict that the trend towards virtual care in svPPA will continue.

## 8. Future Directions in Behavioural Therapies in svPPA

We have shown in this synthesis of evidence that there has been relatively little research on intervention for words and concepts that the person with svPPA can no longer understand, and that therapy gains for such words show limited generalisation. Far from indicating that conceptual restoration or generalisation is not possible, we argue that optimal treatments may not yet have been found, and that this should motivate future research. On the other hand, the use of compensatory approaches to supporting communication and activities of daily living (e.g., assistive technologies) is promising and has the potential to make a difference to the lives of people with svPPA. The next steps should therefore be directed towards: (1) the development of more precise naming therapies, tailored to the level of semantic degradation of the words and concepts treated; (2) finding ways to guarantee transfer and generalisation of therapy gains to daily life; (3) expanding research into the use of assistive technologies, compensatory strategies, programmes to support daily living and how and when to combine these components.

## 9. Conclusions

The last two decades have witnessed rapid advances in the understanding and treatment of svPPA. The current body of research suggests that people with svPPA who have access to non-pharmacological therapies show favourable outcomes and long-lasting effects that can have benefits for health outcomes. Moreover, these treatments are generally well accepted. Although there is a lack of empirical research examining what the optimal combination and timing for treatments are, there are general guidelines for delivering language therapy at different stages of PPA that offer pragmatic advice about how to combine different therapy approaches in a meaningful way [105]. Current ongoing research around the staging of PPA (including svPPA) will make it easier to match therapies to impairments in the future. We therefore advocate for the svPPA care pathway to include a wide range of therapeutic options including both restorative and compensatory strategies and educational and support groups for people with svPPA as well as their care partners. These therapeutic options have the potential to become more accessible due to the advent of telemedicine, which has overcome geographical barriers and can provide care of similar efficacy to face-to-face therapy. Finally, to facilitate dissemination beyond an academic audience we have included a lay summary in English, Spanish and French (Appendix A).

## Figures and Tables

**Figure 1 brainsci-11-01552-f001:**
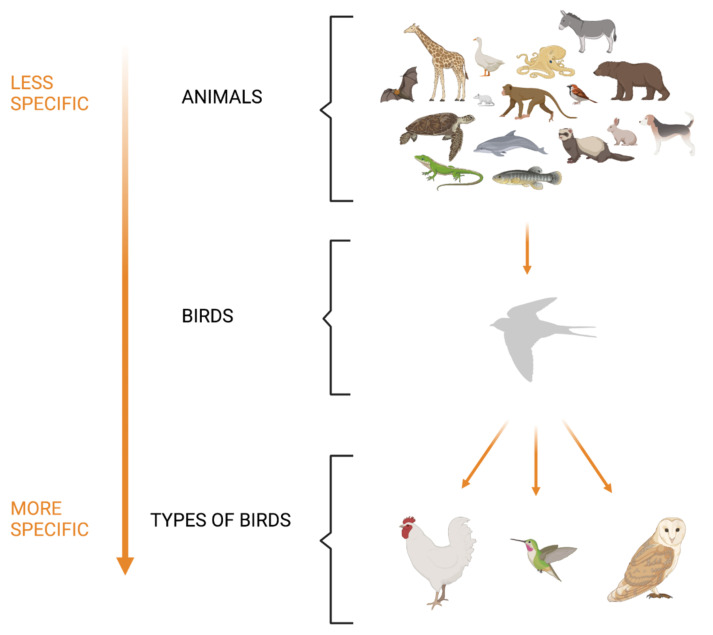
Organisation of the semantic memory category system and its implications for semantic breakdown in svPPA. The characteristic pattern of semantic organisation for the concept “birds” is illustrated in the picture above. Superordinate categories (e.g., animals) sit at the top of the semantic hierarchy. They display a high degree of generality and low specificity among the features shared by their members. Subordinate categories are a more specific level of categorisation, e.g.,” birds” is a subordinate category of “animals” and “hummingbird” is a subordinate category of “birds”. At the bottom of the hierarchy sit the most specific attributes, which are also those to degrade first in svPPA, e.g., “a hummingbird is a very small bird, feeds on flower nectar and can hover”. A typical patient with svPPA may initially name the picture of a hummingbird correctly, but as the disease progresses, errors and superordinate responses would emerge in the following pattern: Assessment 1: Hummingbird → “hummingbird” (named correctly); Assessment 2: Hummingbird → “sparrow” (named as a semantically similar category coordinate); Assessment 3: Hummingbird → “bird” (named as a higher-familiarity typical member of the category); Assessment 4: Hummingbird → “animal” (named as the superordinate category); Assessment 5: Hummingbird → “I don’t know”.

**Figure 2 brainsci-11-01552-f002:**
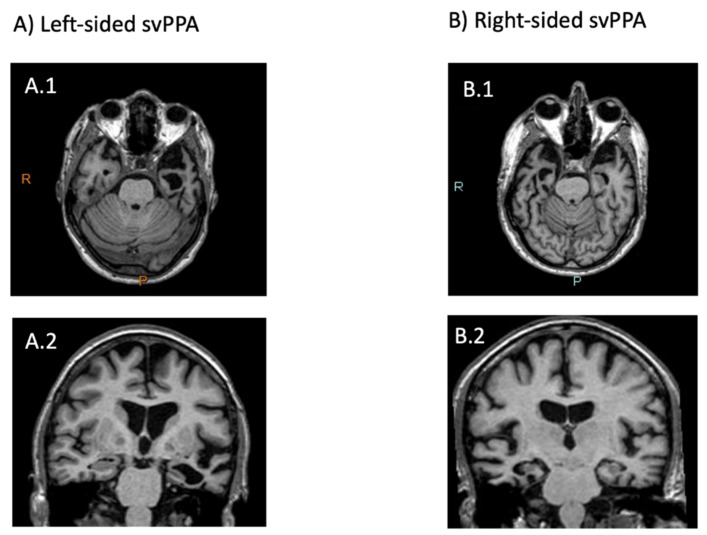
(**A.1**) Axial T1 MR image: anterior temporal lobe displaying marked atrophy on the left pole. (**A.2**) Coronal T1 MR image: marked left temporal atrophy with dilation of the temporal horn and left hippocampal shrinkage. (**B.1**) Axial T1 MR image: anterior temporal lobe displaying bilateral atrophy more marked on the right. (**B.2**) Coronal T1 MR image: marked right temporal atrophy with dilation of the temporal horn and right hippocampal shrinkage.

**Table 1 brainsci-11-01552-t001:** Differences between right and left variant: implications for clinical practice.

	Left-Sided svPPA	Right-Sided svPPA
*Verbal tasks*		
Verbal tasks	Poorer	Better
Single word comprehension	+ impaired	- impaired
Naming	+ impaired	- impaired

*Type of naming errors*		
Circumlocutions	+ frequent	- frequent
omissions	+ frequent	- frequent
Semantic errors	- frequent	+ frequent

*Visual/non-verbal tasks*		
Non-language semantics	Better	Poorer
Prosopagnosia *	- frequent	+ frequent

*Behaviour*		
Social awkwardness	- frequent	+ frequent
Loss of insight	- frequent	+ frequent
Loss of empathy	- frequent	+ frequent
Disinhibition	- frequent	+ frequent
Compulsiveness	- frequent	+ frequent
Apathy	- frequent	+ frequent


(+) means more; (-) means less; * ”prosopagnosia” is a term that refers to impaired ability to recognise faces. It was used by previous authors in the clinical description of the syndrome. It is however worth noting that the recognition deficit seen in right-svPPA is not restricted to faces but encompasses multimodal person knowledge as well. Grey background indicates features more severely impaired or more frequent symptoms in one variant compared to the other.

**Table 2 brainsci-11-01552-t002:** List of lexical retrieval techniques used in the svPPA rehabilitation literature.

Technique	Example
* **Reading and repetition in the presence of a picture** *	Picture presented + corresponding printed word [18,40,52,53,55,58,60,62,63,66,68,69,70,71]
	Picture presented + corresponding printed word + audio recording of the object name (some authors have also included audio recorded descriptions of the treated item and some others also require a written response) [51,54,57,65,67,72]
* **Semantic treatment** *	Picture presented + corresponding spoken + written name + specific attributes [59]
	Semantic feature analysis—this technique requires patients to describe each feature of a word in a systematic way by answering a set of questions about group, use, action, properties, location and association [73,74]
	Conceptual enrichment therapy—this technique manipulates the encoding of new learning to promote flexible learning by placing the trained item in a personally meaningful temporal and spatial context [66,67,70]
	Feature generation from a list of sentence cues for personally relevant episodic or semantic information [75]
	Elaboration of items within subcategories, sorting pictures and words by subcategory, identifying semantic attributes of exemplars, usage of a picture dictionary organised by categories [76]
* **Sentence generation** *	Picture presented + name of the item + example sentence using the word + blank line for the participant to write their own [65]
* **Semantic, phonological, orthographic and/or autobiographical cueing/treatment** *	Sequence of tasks to engage semantic, phonemic, and orthographic self-cues and/or autobiographic memories, e.g., prompt semantic description by asking “what do you use it for?” [56,62,64,77,78,79,80,81,82,83]

Note: This is not intended to be a systematic review of naming therapy techniques. It rather aims to offer a practical overview of commonly administered training strategies. See [66] for a review of methods used in svPPA studies up until 2014 and [23] for methods used in PPA studies in general.

## Data Availability

Not applicable.

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
