# Peer review of "Semantic Variant Primary Progressive Aphasia: Practical Recommendations for Treatment from 20 Years of Behavioural Research"

_brainsci, 2021, doi:10.3390/brainsci11121552_

Round 1

Reviewer 1 Report

The present paper offers a very interesting and exhaustive revision of the literature on the treatment of svPPA, adding useful practical recommendations.

The content is clear and well written and it is very pleasant to read.

I only suggest to remove duplicate references (e.g. ref 69 &70, 61 &74, 97 &53). 

Author Response

On behalf of my co-authors, I would like to thank the three reviewers for their time and work dedicated to our manuscript and for their constructive and thorough feedback. We believe their comments have substantially improved the quality of the piece. We have addressed every point below.

Reviewer 1

The present paper offers a very interesting and exhaustive revision of the literature on the treatment of svPPA, adding useful practical recommendations. The content is clear and well written and it is very pleasant to read. I only suggest to remove duplicate references (e.g. ref 69 &70, 61 &74, 97 &53).

#Response:

Thank you for this encouraging comment and for identifying the duplicate references. The duplicate references have now been removed.

Reviewer 2 Report

This study aimed to provide an overview of the last two decades of research in language and behavioural therapies in svPPA. The article is written by expert clinicians and researchers in this field. The article is comprehensive, interesting, and a relevant resource for clinicians interested in svPPA and language rehabilitation. The article is well-written. Nevertheless, I have some minor comments and questions, listed below:

Comment 1a. The Conclusions section is lengthy. I’m wondering if the authors should create an additional section before the Conclusions that discusses ‘8. Future directions in language therapies in svPPA’. Aspects from the Conclusions could be included in this section. Also, this new section might provide a good opportunity for the authors to discuss what they are currently doing in the area of language rehabilitation. For example, are the authors working with industry partners to create a personalised svPPA word retraining app?

Comment 1b. I’m also wondering if the Appendix 1 Supplementary Material could be referred to elsewhere/earlier in the manuscript? This material is referenced in the abstract but is only referred to in the closing statements of the manuscript. If the authors decide to refer to the Appendix 1 material in the Conclusions section, then should it remain an important point in the abstract?

Comment 2. There are a few grammatical, spelling, and formatting errors throughout the paper. The authors should consider using a grammar check software such as Grammarly. Below is a list of potential errors and wording suggestions that I noted down whilst reading the manuscript.

  • There is inconsistency in the use of British and North American English. For example, the authors use the terms ‘behavioural’ and ‘behavioral’ interchangeably. Other examples include ‘generalisation’, ‘recognize’ etc. Please consider using British or North American English.
  • The authors have submitted a manuscript that includes suggested track changes that have not been dealt with—for example, lines 104, 171 etc.
  • In some instances, citations are together, e.g., (3,5), whereas in other instances, citations are separate, e.g., (3)(4).
  • The authors use e.g. and e.g., interchangeably.
  • Lines 26 and 27 — The emphasis of this article is on language interventions. Consider changing the sentence on lines 26 and 27 to: ‘This article offers an overview of the last two decades of research in language and behavioural therapies in svPPA….’
  • Lines 32-33 — There is a typo in the abstract: developedment
  • Line 58 — Did the authors mean to use the term ‘established’ or ‘stablished’?
  • Table 1. I am unsure why some of the words are highlighted in grey whilst others are not.
  • Line 220 — The authors should qualify the sentence ‘Benefits of therapy have been shown in individuals with svPPA regardless of the severity of anomia.’ with relevant citations. Also, I wonder if treatments have been shown to be effective in svPPA patients who are mute? If this has not been shown, then consider revising this sentence.
  • Line 368 — Should it be ‘What evidence do we have?’
  • Line 374. There should be a full stop between ‘,113) Ro-‘

Author Response

On behalf of my co-authors, I would like to thank the three reviewers for their time and work dedicated to our manuscript and for their constructive and thorough feedback. We believe their comments have substantially improved the quality of the piece. We have addressed every point below.

Reviewer 2

This study aimed to provide an overview of the last two decades of research in language and behavioural therapies in svPPA. The article is written by expert clinicians and researchers in this field. The article is comprehensive, interesting, and a relevant resource for clinicians interested in svPPA and language rehabilitation. The article is well-written. Nevertheless, I have some minor comments and questions, listed below:

Comment 1a. The Conclusions section is lengthy. I’m wondering if the authors should create an additional section before the Conclusions that discusses ‘8. Future directions in language therapies in svPPA’. Aspects from the Conclusions could be included in this section. Also, this new section might provide a good opportunity for the authors to discuss what they are currently doing in the area of language rehabilitation. For example, are the authors working with industry partners to create a personalised svPPA word retraining app?

#Response:

Thank you for this observation. We have added the suggested section that now reads: “We have shown in this synthesis of evidence that there has been relatively little research on intervention for words and concepts that the person with svPPA can no longer understand, and that therapy gains for such words show limited generalisation. Far from indicating that conceptual restoration or generalisation is not possible, we argue that optimal treatments may not yet have been found, and that this should motivate future research. On the other hand, the use of compensatory approaches to supporting communication and activities of daily living (e.g., assistive technologies) is promising and has the potential to make a difference to the lives of people with svPPA. The next steps should therefore be directed towards 1) the development of more precise naming therapies, tailored to the level of semantic degradation of the words and concepts treated; 2) finding ways to guarantee transfer and generalisation of therapy gains to daily life; and 3) expanding research into the use of assistive technologies, compensatory strategies, programmes to support daily living and how and when to combine these components.”

Comment 1b. I’m also wondering if the Appendix 1 Supplementary Material could be referred to elsewhere/earlier in the manuscript? This material is referenced in the abstract but is only referred to in the closing statements of the manuscript. If the authors decide to refer to the Appendix 1 material in the Conclusions section, then should it remain an important point in the abstract?

#Response:

We agree with this comment. We have inserted it at the end of the introduction.

Comment 2. There are a few grammatical, spelling, and formatting errors throughout the paper. The authors should consider using a grammar check software such as Grammarly. Below is a list of potential errors and wording suggestions that I noted down whilst reading the manuscript.

There is inconsistency in the use of British and North American English. For example, the authors use the terms ‘behavioural’ and ‘behavioral’ interchangeably. Other examples include ‘generalisation’, ‘recognize’ etc. Please consider using British or North American English.

#Response:

Thank you for this remark. We have now reviewed the document using British English and kept it consistent across the text.

The authors have submitted a manuscript that includes suggested track changes that have not been dealt with—for example, lines 104, 171 etc.

#Response:

We apologise for this mistake. We have carefully revised the new version and we hope the new submission is free from unaddressed tracked changes.

In some instances, citations are together, e.g., (3,5), whereas in other instances, citations are separate, e.g., (3)(4). The authors use e.g. and e.g., interchangeably.

#Response:

Thank you for this comment. We now use “e.g.,” consistently across the manuscript and amended the referencing style.

Lines 26 and 27 — The emphasis of this article is on language interventions. Consider changing the sentence on lines 26 and 27 to: ‘This article offers an overview of the last two decades of research in language and behavioural therapies in svPPA....’

#Response:

Thank you for the suggestion. We have amended this sentence that now reads “This article offers an overview of the last two decades of research into interventions to support in people with svPPA …”

Lines 32-33 — There is a typo in the abstract: developedment

#Response:

This typo has been corrected.

Line 58 — Did the authors mean to use the term ‘established’ or ‘stablished’?

#Response:

This typo has been corrected and the word replaced by “established”

Table 1. I am unsure why some of the words are highlighted in grey whilst others are not.

#Response:

We apologise for the lack of clarity of this code. It meant to highlight the differences in profile between left- and right- variants but we agree it looks confusing for readers as it currently stands. We have now added a footnote description in Table 1 that reads “Highlights in grey indicate features more severely impaired or more frequent in one variant compared to the other.”

Line 220 — The authors should qualify the sentence ‘Benefits of therapy have been shown in individuals with svPPA regardless of the severity of anomia.’ with relevant citations. Also, I wonder if treatments have been shown to be effective in svPPA patients who are mute? If this has not been shown, then consider revising this sentence.

#Response:

We agree with this remark. The new sentence reads “Benefits of therapy have been shown in individuals with svPPA across a range of anomia severities but with some preservation of spoken language (no studies have currently been reported with mute individuals)”. References have been added.

Line 368 — Should it be ‘What evidence do we have?’

#Response:

This change has been added. The new sentence reads “What evidence do we have about prophylactic treatment in svPPA?”

Line 374. There should be a full stop between ‘,113) Ro-‘

#Response:

The full stop has been added.

Reviewer 3 Report

Please refer to the attached document, thank you!

Author Response

On behalf of my co-authors, I would like to thank the three reviewers for their time and work dedicated to our manuscript and for their constructive and thorough feedback. We believe their comments have substantially improved the quality of the piece. We have addressed every point below.

Reviewer 3
Manuscript Title: Semantic Variant Primary Progressive Aphasia: practical recommendations for treatment from 20 years of behavioral research
Authors: Aida Suárez-González, Sharon A Savage, Nathalie Bier, Maya L. Henry, Regina Jokel, Lyndsey Nickels and Cathleen Taylor-Rubin.

Article Summary: The authors provide an overview of non-pharmacological treatment options for patients with Semantic Variant Primary Progressive Aphasia (svPPA), including behavioral and language therapies. Additionally, clinical recommendations and directions for future research are provided. The authors discuss left versus right variants of svPPA and how the therapies for each differ. They also provide an overview of maintenance and generalization therapy gains in svPPA patients in research studies thus far. Impairment-based treatment for anomia and word comprehension, compensatory strategies, interventions to support activities of daily living, and support groups and educational programs are discussed, and an integrative approach of these treatments is recommended.

Strengths: The authors provide a thorough overview of treatments for svPPA patients. They are clear in differentiating left versus right variants of svPPA and the implications for clinical practice. The section on impairment-based treatment of anomia and word comprehension deficits is well structured and easy to understand by readers. The authors also explain the importance of support groups and educational programs and the elements the groups should include. There are no major weaknesses. The minor weaknesses are listed below.

#Response:

Thank you for these supportive and encouraging comments.

Minor weaknesses:

Section 1: Introduction

First paragraph (page 1-2, lines 3-4): Semantic dementia and svPPA are not always referred to as the same diagnosis.

#Response:

Thank you for this remark. We understand that there may be some who feel that they are distinct. However, it is our belief (including personal communication with some of the key authors who used the older term, e.g., John Hodges) that the majority of people now use the terms interchangeably. We do not intend to enter this debate in the paper but aim to reflect in the text that different terms have been used, particularly in different time periods, to avoid confusion among readers who may find different labels in different texts. The key issue is that the major language features of semantic dementia and svPPA are the same.

Table 1
o Is there a way to indicate “more” versus “less” or “more impaired” versus “less impaired” in a simpler way? Symbols? The language used in comparing characteristics between left-sided and right-sided svPPA takes away from the information in the table. Since the definition of prosopagnosia is defined earlier, it seems unnecessary to place it under the table, but rather elaborate more in the previous paragraph.

#Response:

Thank you for this comment. We agree the current layout of the table is repetitive. We have now used ‘+’ to indicate that a feature is more frequent/pronounced, and ‘-‘ to indicate that a feature is less frequent/pronounced. Regarding the term prosopagnosia, the reason for not elaborating more in the text is to improve readability. Although a lengthy explanation about the nature of prosopagnosia in svPPA is not within the scope of this article, we are aware the term may not be clear for all readers, hence the footnote in the table.

Section 4: The current evidence informing treatment and management of anomia and word comprehension deficits. In the paragraph “Who benefits from anomia therapy?” are you saying that severely impaired svPPA patients would benefit from impairment-based treatments? Would you say you recommend this type of treatment over compensatory strategies or in conjunction? How much should the caregiver be involved? It would be helpful to clarify.

#Response:

Thank you for this helpful observation. We do not have available evidence reporting the effects of therapy on mute individuals and have amended the sentence accordingly. Unfortunately, there is no evidence to allow the recommendation of one type of therapy over the other since no comparative studies have been produced in this regard to date. Neither have there been studies examining the level of involvement of caregivers, as far as we know.  Nevertheless, a pragmatic approach that combines impairment-based and compensatory strategies seem in line with recent expert clinical advice (Jokel, 2021). We have referenced it on the conclusions (see also response to your question regarding patients with moderate-severe impairment two paragraphs below).

Section 5: Compensatory strategies to support communication in svPPA
o Have there been any studies looking at compensatory strategies not involving AAC? The reader may anticipate learning about how strategies such as purposefully circumlocuting or structured cueing from caregivers may improve communication for svPPA patients.

#Response:

This is indeed a very important point. Strategies of functional communication have been explored in individuals with PPA in general, with mixed results regarding efficacy. No studies examining non-AAC compensatory strategies specifically targeting svPPA have been identified by this research team. We have added the following line to the end of section 5: “Finally, although strategies of functional communication have been explored in individuals with PPA in general, there is a lack of studies examining non-AAC compensatory strategies targeting svPPA.”

Since the emphasis of this paper is impairment-based treatment rather than compensatory, would you say that moderate-severe patients should try that type of treatment first or both at the same time? It would be helpful to add a paragraph about what is recommended after reviewing the treatment options for different severities of patients.

#Response: This piece aims to provide recommendations based on the best available evidence. What this reviewer poses is a challenge that we all face in clinics every day and that it is of great relevance. We have added the following paragraph to the conclusions section: “Although there is a lack of empirical research examining what the optimal combination and timing for treatments are, there are general guidelines in delivering language therapy at different stages of PPA offering pragmatic advice about how to combine different therapy approaches in a meaningful way (Jokel, 2021). Current ongoing research around the staging of PPA (including svPPA) will make it easier to match therapies to impairments in the future.”

Section 8: Conclusions
o Recommendations for clinical practice on how best to combine the different treatments (impairment-based treatment for anomia and word comprehension, compensatory strategies, interventions to support ADLs, support groups/educational programs) should be added.

#Response:

Thank you for this remark. Although we cannot provide evidence-based recommendations on how to best combine the different treatments, we agree we should at least advocate for people with svPPA being offered a wide range of therapeutic options that are already proving promised. This has been added to the Conclusions section along with a reference to Jokel’s (2021) recommendations.
